# Risk Factors for Delirium after Deep Brain Stimulation Surgery under Total Intravenous Anesthesia in Parkinson’s Disease Patients

**DOI:** 10.3390/brainsci13010025

**Published:** 2022-12-22

**Authors:** Wenbin Lu, Xinning Chang, Lulong Bo, Yiqing Qiu, Mingyang Zhang, Jiali Wang, Xi Wu, Xiya Yu

**Affiliations:** 1Faculty of Anesthesiology, Changhai Hospital, Naval Military Medical University, Shanghai 200433, China; 2Department of Neurosurgery, Changhai Hospital, Naval Military Medical University, Shanghai 200433, China; 3Department of Chemistry, University of Utah, 201 Presidents’ Cir, Salt Lake City, UT 84112, USA; 4Department of Anesthesiology and Perioperative Medicine, Shanghai Fourth People’s Hospital, School of Medicine, Tongji University, Shanghai 200434, China; 5Shanghai Key Laboratory of Anesthesiology and Brain Functional Modulation, Shanghai 200434, China; 6Translational Research Institute of Brain and Brain-Like Intelligence, Shanghai Fourth People’s Hospital, School of Medicine, Tongji University, Shanghai 200434, China; 7Clinical Research Center for Anesthesiology and Perioperative Medicine, Tongji University, Shanghai 200434, China

**Keywords:** deep brain stimulation, general anesthesia, Parkinson’s disease, postoperative delirium, risk factors, total intravenous anesthesia

## Abstract

Background: Postoperative delirium (POD) is associated with perioperative complications and mortality. Data on the risk factors for delirium after subthalamic nucleus deep brain stimulation (STN-DBS) surgery is not clarified in Parkinson’s disease (PD) patients receiving total intravenous anesthesia. We aimed to investigate the risk factors for delirium after STN-DBS surgery in PD patients. Methods:The retrospective cohort study was conducted, including 131 PD patients who underwent STN-DBS for the first time under total intravenous anesthesia from January to December 2021. Delirium assessments were performed twice daily for 7 days after surgery or until hospital discharge using the confusion assessment method for the intensive care unit. Multivariate logistic regression analysis was used to determine the risk factor of POD. Results: In total, 22 (16.8%) of 131 patients were in the POD group, while the other 109 patients were in the Non-POD group. Multivariate logistic regression analysis showed that preoperative Mini-mental State Examination score [odds ratio = 0.855, 95% confidence interval = 0.768–0.951, *p* = 0.004] and unified Parkinson’s disease rating scale part 3 (on state) score (odds ratio = 1.061, 95% confidence interval = 1.02–1.104, *p* = 0.003) were independently associated with delirium after surgery. Conclusions: In this retrospective cohort study of PD patients, a lower Mini-mental State Examination score and a higher unified Parkinson’s disease rating scale part 3 (on state) score were the independent risk factors for delirium after STN-DBS surgery in PD patients under total intravenous anesthesia.

## 1. Introduction

Postoperative delirium (POD) is one of the most common postoperative complications in elderly patients, which is characterized by fluctuation in consciousness over time and impaired concentration. The incidence of POD in different surgery ranges from 5% to 50% [1,2]. Parkinson’s disease (PD) is one of the most common progressive neurodegeneration diseases. Given the side effects of drug therapy and disease progression, subthalamic nucleus deep brain stimulation (STN-DBS) is the most effective treatment method for advanced PD patients [3,4,5,6]. Previous studies have shown that PD patients often experienced neuropsychiatric symptoms, of which POD was the most common symptom with an incidence of 22–42.6% [5,7]. POD not only exacerbated the overall prognosis of PD patients and increased the risk of cognitive impairment, but also increased mortality [8].

Numerous studies have shown that age, preoperative cognitive impairment, preoperative nutritional status, electrolyte disturbance, operation time, and postoperative brain edema were associated with POD in PD patients [9,10,11]. The results were controversial due to the different anesthesia methods in the studies. Furthermore, a recent study has also shown that compared with inhalation anesthesia, the incidence of POD is lower in elderly patients with total intravenous anesthesia [12]. However, the risk factor for delirium after STN-DBS surgery in PD patients under total intravenous anesthesia has yet to be investigated.

Therefore, the purpose of this study is to investigate the risk factor for delirium after STN-DBS surgery in PD patients under total intravenous anesthesia. For this aim, the present study was conducted to provide new insight into the early recognition and perioperative management of delirium after STN-DBS surgery in PD patients.

## 2. Materials and Methods

### 2.1. Study Setting and Design

We conducted a retrospective cohort study of PD patients who underwent elective STN-DBS surgery for the first time under total intravenous anesthesia in the Faculty of Anesthesiology of Changhai Hospital. From the DoCare anesthesia clinical information system (MedTech, V3.1.0build153) and Jiahe electronic medical record system, we retrieved and obtained 173 patients who underwent DBS in our hospital from January to December of 2021. This study was approved by the ethics committee of our hospital (CHEC2020-151) and published in clinicaltrials.gov (NCT04696978). Written informed consent was obtained from all patients.

### 2.2. Participants

This retrospective cohort study included patients who received elective STN-DBS surgery for the first time under total intravenous anesthesia. Inclusion criteria included: total intravenous anesthesia; DBS surgery (bilateral, STN); age ≥ 55 years; and complete patient information. Exclusion criteria were as follows: American Society of Anesthesiologists (ASA) physical status > III; preoperative neuropsychiatric symptoms; preoperative cognitive decline; intraoperative severe cardiovascular events (such as cardiac arrest); secondary surgery; and a history of alcoholism.

### 2.3. Assessment of Delirium

POD was defined as an acute state characterized by time-fluctuating altered consciousness and inattention that occurred in the hospital within 1 week after surgery or before discharge and met the DSM-V diagnostic criteria. The confusion assessment method for the intensive care unit (CAM-ICU) was used to evaluate POD due to intracranial operation in patients who were admitted to the neurosurgery ICU (NICU) after surgery. Delirium assessments were done twice daily for 7 days after surgery or until hospital discharge. The morning and evening assessments were made before 10 AM and after 5 PM. The CAM-ICU questionnaire was performed by a qualified doctor who was trained with the CAM-ICU training manual [13].

### 2.4. Surgical Procedure and Anesthesia Method

The surgical procedure was consistent with our previous study [14]. Patients were given standard monitoring after entering the operating room, including non-invasive blood pressure (BP), heart rate (HR), pulse oximetry saturation (SpO_2_), and end-tidal carbon dioxide partial pressure (P_ET_CO_2_). After the patient’s peripheral venous access was established, dexamethasone 8mg, sufentanil 0.3–0.5 µg kg^−1^, propofol 2–3 mg kg^−1^, and rocuronium 0.6 mg kg^−1^ were given for the induction of anesthesia. After tracheal intubation, P_ET_CO_2_ was maintained at 35–45 mmHg by mechanical ventilation in a volume-controlled mode with a tidal volume of 6–8 mL kg^−1^ and a respiratory rate of 12–15 breaths per minute. Anesthesia was maintained by continuous infusion of propofol 6–8 mg kg^−1^ h^−1^ and remifentanil 0.1–0.2 µg kg^−1^ min^−1^ after induction of anesthesia to maintain stable hemodynamics. Sufentanil 10 µg was administrated half an hour before the end of the operation, and the infusion of propofol and remifentanil was stopped after the operation. The patients were then transferred to the post-anesthesia care unit (PACU), where the tracheal tube was to be removed. When the Aldrete score was ≥9, patients were transferred to NICU for the first day after surgery and then transferred to the surgical ward.

### 2.5. Data Collection

We collected general patient information on age, gender, body mass index (BMI), ASA physical classification, education level, length of hospital stay, medical history (hypertension, diabetes, coronary heart disease), operation duration, PD duration, family history of PD, physiological parameters (albumin, sodium, potassium, chloride and glucose) and visual analog scale (VAS) pain score after surgery as well as perioperative MMSE score and postoperative hallucination. The PD-related symptoms were also recorded as follows: dopamine equivalent dose, preoperative motor symptoms (rigid retardation, tremor, frozen gait, balance dysfunction, weakness), non-motor symptom scale (NMSS), movement disorder society-unified Parkinson’s disease rating scale (MDS-UPDRS), unified dyskinesia rating scale (UDysRs), standardized swallowing assessment (SSA), Kubota drinking test, KINGS Parkinson’s disease pain scale (KPPS), Hamilton Depression Scale (HAMD), and Hamilton Anxiety Scale (HAMA).

In addition, data on anesthesia and brain CT imaging 24 h after surgery was obtained from an electronic medical information system, including the dose of anesthesia drugs (propofol, sufentanil, remifentanil, rocuronium), brain hemorrhage, edema and pneumocephalus after surgery.

### 2.6. Statistical Analysis

Statistical analysis of the data was performed using SAS 9.4 statistical software (SAS Institute, Cary, NC, USA). Continuous variables in normal distribution expressed as mean ± standard deviation were compared using Student’s *t*-test, while those in non-normal distribution expressed as median (IQR) were compared using the Mann-Whitney U-test. Categorical variables represented as frequency and percentages were compared using a chi-square test or Fisher’s exact test. Logistic regression analysis was used to evaluate the risk factor of postoperative delirium.

Considering the total number of POD (*n* = 22) in our study and to avoid overfitting in the model, two variables were chosen for multivariate analysis on the basis of previous findings and clinical constraints. A previous study has shown that preoperative MMSE score was related to delirium after surgery [15]. In addition, the UPDRS part 3 score was associated with delirium after DBS surgery and short-term motor outcome in PD patients [16,17]. Thus, we chose the preoperative MMSE score and UPDRS part 3 (on state) score as the two variables for our multivariable logistic regression model.

We excluded variables from the univariable analysis if they had collinearity with the preoperative MMSE score. Furthermore, we also conducted a sensitivity analysis controlling for variables with *p* < 0.2 from the univariable analysis to evaluate the risk factor of postoperative delirium. A *p* < 0.01 for multiple comparisons between two groups and *p* < 0.05 for other analyses were considered statistically significant.

## 3. Results

From January to December of 2021, a total of 173 patients were collected, and 42 were excluded (26 were younger than 55 years old, 15 had a history of DBS, and one received globus pallidus internus DBS). We included 131 patients in the final analysis, where 22 patients (16.8%) developed POD, including 15 males (68.2%) and 7 females (31.8%). Figure 1 shows a participant flow diagram.

### 3.1. General Patient Information

General patient information is listed in Table 1. Compared with the Non-POD group, the patients in the POD group were significantly older (*p* < 0.001) and had a higher incidence of diabetes (*p* < 0.05). Furthermore, MMSE scores at 24 h, 72 h, and 1 month after surgery were significantly lower in patients with delirium (*p* < 0.01). However, other general patient information between the two groups was not statistically significant.

### 3.2. Parkinson’s Motor Symptoms and Non-Motor Symptoms

Compared with the Non-POD group, the POD group had a higher NMSS score (*p* < 0.05), higher SSA score (*p* < 0.01), and higher UPDRS part 1, 2, 3 (on/off state) score (*p* < 0.05) before surgery. Other symptoms listed in Table 2 between the two groups were not statistically significant.

### 3.3. Anesthetic and Postoperative Brain Imaging Data

The postoperative brain CT imaging data listed in Table 3 indicated patients with delirium had a higher incidence of intracranial hemorrhage (*p* < 0.001) and brain edema (*p* < 0.05). Anesthetic data between the two groups was not statistically significant in Table 3.

### 3.4. Risk Factors for Delirium after STN-DBS Surgery

Outcomes of the multivariate logistic regression analysis shown in Table 4 indicated that the preoperative MMSE score (OR = 0.855, 95% CI = 0.768–0.951, *p* = 0.004) and UPDRS part 3 (on state) score (OR = 1.061, 95% CI = 1.02–1.104, *p* = 0.003) were independently associated with delirium after STN-DBS surgery under total intravenous anesthesia. When adjusting for variables with *p* < 0.2 from the univariable analysis, we showed similar results (Appendix A). In addition, the area under the receiver operating characteristic curve for preoperative MMSE score combined with UPDRS part 3 (on state) score as a predictor of POD was 0.784 (0.675–0.893) (Figure 2).

## 4. Discussion

STN-DBS is the most effective method for treating advanced PD, which can reduce non-motor symptoms caused by high-dose dopamine treatment due to reducing dopamine drug treatment [18,19]. However, previous studies have shown that neuropsychiatric symptoms often occurred after STN-DBS surgery, resulting in delirium with an incidence of 22–42.6% [5,7]. This study showed that the incidence of POD was 16.8% in patients receiving STN-DBS under total intravenous anesthesia, which was lower than in previous studies. This may be associated with patients receiving total intravenous anesthesia rather than inhalation anesthesia. In addition, perioperative anti-delirium strategies are now routine in patients receiving surgery, which may reduce the risk of delirium after surgery. As POD exacerbated the overall prognosis and cognitive function in PD patients, it is essential to identify high-risk patients. To our knowledge, this is the first retrospective study to investigate risk factors for delirium after STN-DBS surgery under total intravenous anesthesia. We found that a preoperative lower MMSE score and higher UPDRS part 3 (on state) score were the independent risk factors for delirium after STN-DBS surgery in PD patients under total intravenous anesthesia.

The risk of POD increases significantly in elderly patients, which was related to a poor basic condition and lower preoperative cognitive function reserve [20,21,22]. Consistent with previous literature, we found that older age, higher comorbidity (especially diabetes) burden, and lower MMSE scores were associated with delirium after STN-DBS surgery. We also showed that preoperative electrolyte level (especially serum sodium and serum chloride) was low, and serum glucose level was high in patients with POD, which is similar to previous studies [23,24,25]. In addition, the VAS pain score after surgery was high in patients with delirium. However, the difference in electrolytes, blood glucose values, and VAS pain scores between the two groups is very small and not clinically meaningful. More importantly, patients with delirium had poor PD-related symptoms before surgery and a higher incidence of brain hemorrhage and edema after surgery.

In parallel with our results, aging and diabetes are involved in oxidative stress which can contribute to delirium [26]. The previous study has reported that disturbances in brain neuronal networks due to electrolyte disorder and higher serum glucose level may be a cause of delirium [27]. In addition, hyperglycemia is a response to inflammation and is followed by neuro-inflammation, and insulin has an anti-inflammatory effect [28]. Therefore, patients with diabetes and hyperglycemia may possess baseline inflammatory conditions which result in delirium.

The preoperative MMSE scores of the patients collected in this study were all within the normal range. The cut-off of the MMSE score referred to the previous study in China [29]. Therefore, a preoperative MMSE score above 16 for illiterate patients, above 19 for patients with 1–6 education years or above 23 for patients with more than 6 education years was the indication for DBS in PD patients in the study. However, we found that patients with a preoperative low MMSE score were susceptible to suffering from delirium after surgery. In particular, Parkinson’s patients have poor cognitive reserve due to pathological changes and inflammation in the central nervous system [30], which can contribute to delirium after surgery. In addition, the present study showed that patients with POD tended to suffer from a lower MMSE score after surgery. This indicated that delirium may be associated with a decline in cognitive ability after surgery, which is consistent with the previous study [31]. Logistic regression analysis further showed that a low preoperative MMSE score was an independent risk factor for POD.

Previous studies have suggested that PD-related motor symptoms and non-motor symptoms were associated with POD [32,33]. In line with previous studies, we found that patients with POD were more likely to have poor scores on UPDRS parts 1, 2, 3 and NMSS as well as SSA. Parkinson’s disease is associated with α-synuclein aggregation and the death of dopaminergic neurons which result in motor symptoms and non-motor symptoms [34]. Therefore, poor UPDRS, NMSS and SSA scores may indicate serious pathological changes and transmitters disorder in the central nervous system in PD patients, which may contribute to delirium after surgery. Furthermore, we found that the SSA score was associated with POD, which has not been reported in previous studies. Logistic regression analysis further showed that a high UPDRS part 3 (on state) score was also an independent risk factor for POD. Late-onset PD patients with a higher UPDRS part 3 score in the on-state showed that PD-related symptoms were not sensitive to drug treatment. Meanwhile, this may suggest the deterioration of pathological changes and a transmitters disorder in PD patients or the occurrence of drug-induced complications, which may contribute to delirium after surgery.

The previous study has shown that postoperative complications were related to POD [17]. We collected the postoperative brain imaging data and found that brain hemorrhage and edema at the site of stimulation were associated with POD patients. All of them were treated with conservative medical treatments due to mild brain hemorrhage with less than 3ml and mild brain edema. The haemorrhage rate was higher than in previous studies due to multiple postoperative CT examinations in our study. Therefore, surgeons should pay attention to delayed hemorrhage after surgery.

The present study also has some unavoidable limitations. Firstly, this is a small-sample single-center retrospective study, which may result in bias. Secondly, the impact of STN targets on cognition is mainly manifested in complex cognition, and it may be more sensitive to use a variety of cognitive scoring scales than the MMSE score in the present study. Thirdly, only two variables, the preoperative MMSE score and UPDRS part 3 (on state) score, were selected for multivariate logistic regression analysis to avoid overfitting of regression analysis. However, sensitivity analysis controlling for other variables showed similar results. Fourthly, we did not record the duration of postoperative delirium, which could reflect the progression of the disease. Lastly, PD patients often suffer from neuropsychiatry symptoms after surgery, especially hallucinations which may happen earlier than impaired cognition. Although we also observed hallucinations in PD patients after surgery, the relationship between hallucinations and POD will be verified in future studies due to the small number of patients with hallucinations in our study.

## 5. Conclusions

In conclusion, we found that a low preoperative MMSE score and high UPDRS part 3 (on state) score are independent risk factors for delirium after STN-DBS surgery under total intravenous anesthesia in Parkinson’s disease. While further prospective studies need to be performed in order to establish the generalizability of these findings.

## Figures and Tables

**Figure 1 brainsci-13-00025-f001:**
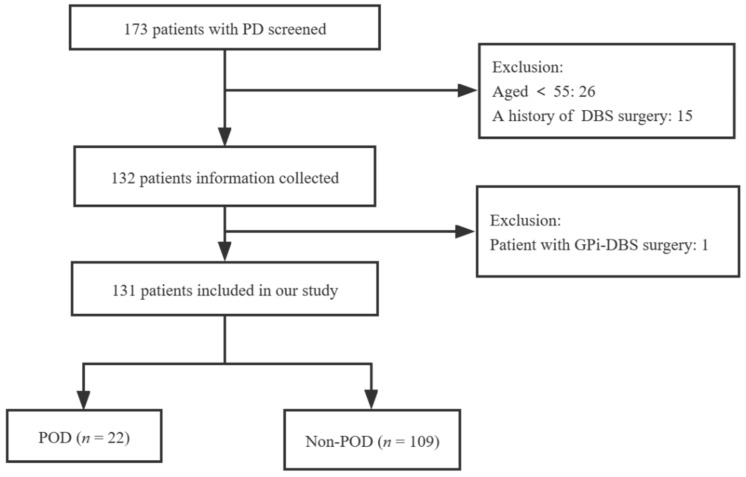
Flow chart of participants in the study.

**Figure 2 brainsci-13-00025-f002:**
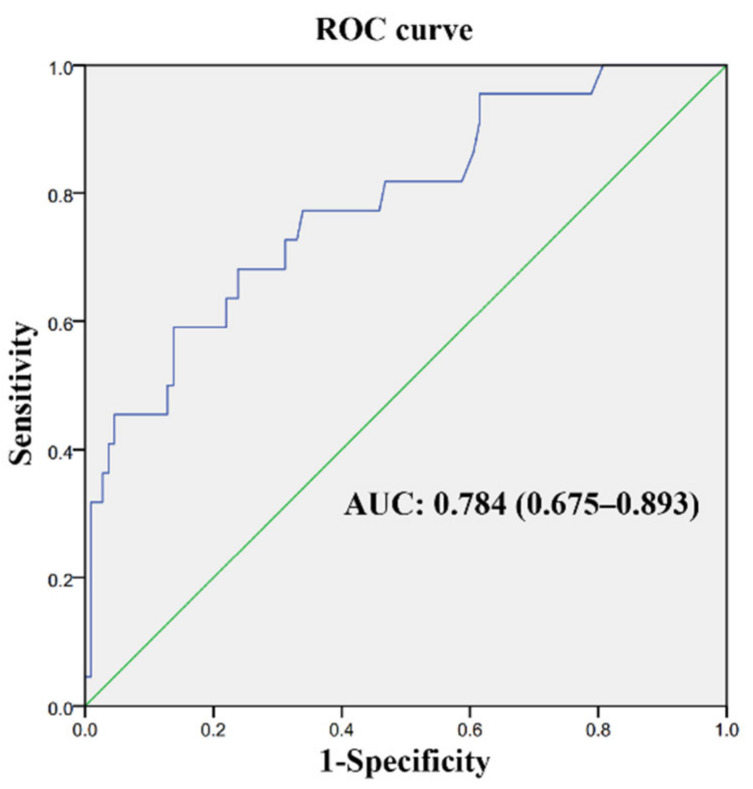
Receiver operating characteristic (ROC) curve for preoperative MMSE score combined with UPDRS part 3 (on state) score as a predictor of POD.

**Table 1 brainsci-13-00025-t001:** Baseline characteristics of the study population.

	POD Group (*n* = 22)	Non-POD Group (*n* = 109)	*p*
Age (years)	68.27 ± 6.46	64.25 ± 5.65	<0.01
Male	15 (68.2)	56 (51.4)	0.15
BMI (kg/m^2^)	24.19 ± 3.65	22.88 ± 3.48	0.11
ASA (II/III)	11/11	73/36	0.13
Education level			0.47
Illiterate school	2	12	
Primary school	5	17	
Middle school	12	52	
Technical secondary school or more	3	28	
Length of hospital stay (days)	9 (7–10)	8 (7–9)	0.18
Hypertension	7 (31.8)	29 (26.6)	0.62
Diabetes	5 (22.7)	6 (5.5)	0.03
Coronary heart disease	0	0	1
Operation duration (min)	135 (130–145)	130 (120–140)	0.09
Family history of Parkinson’s disease	1 (4.5)	20 (18.3)	0.20
Disease duration (years)	9.5 (8–12)	9 (6–12)	0.66
VAS pain score			
Postoperative 24 h	3 (2–4)	2 (1–3)	0.02
Postoperative 72 h	2 (1–3)	2 (1–2)	0.23
MMSE score			
Preoperative	23 (19–28)	27 (26–29)	<0.01
Postoperative 24 h	18 (11–23)	26 (23–28)	<0.0001
Postoperative 72 h	18 (13–23)	26 (23–28)	<0.0001
Postoperative 1 month	22 (17–24)	27 (25–28)	<0.0001
Postoperative hallucination	1	2	0.427
Preoperative serum albumin (g/dl)	44 (42–45)	43 (42–45)	0.69
Serum Na (mmol/L)			
Preoperative	143 (142–144)	144 (142–145)	0.04
Postoperative 24 h	142 (140–143)	142 (141–144)	0.20
Serum Cl (mmol/L)			
Preoperative	105 (103–106)	106 (104–107)	0.04
Postoperative 24 h	106 (105–108)	106 (104–108)	0.95
Serum K (mmol/L)			
Preoperative	3.8 (3.7–4.2)	3.9 (3.7–4.1)	0.56
Postoperative 24 h	3.9 (3.7–4.1)	3.9 (3.8–4.1)	0.63
Serum glucose (mmol/L)			
Preoperative	5.4 (5.3–6.3)	5.3 (4.9–5.7)	0.03
Postoperative 24 h	5.8 (5.5–6.6)	5.6 (5.2–6.4)	0.14

Data are reported as mean ± standard, frequency (percentage), or median (inter-quartile range). BMI, body mass index; ASA, American Society of Anesthesiologists; VAS, visual analogue scale; MMSE, Mini-Mental State Examination; POD, postoperative delirium.

**Table 2 brainsci-13-00025-t002:** Parkinson’s disease related motor and non-motor symptoms.

	POD Group (*n* = 22)	Non-POD Group (*n* = 109)	*p*
Levodopa equivalent daily dose (mg/d)	641.67 (550–850)	675 (520.84–900)	0.90
Rigidity	22 (100)	109 (100)	1
Tremor	20 (90.9)	94 (86.2)	0.81
Freezing of gait	11 (50)	54 (49.5)	0.81
Postural reflex impairment	20 (90.9)	104 (95.4)	0.74
Muscle weakness	22 (100)	109 (100)	1
UPDRS part 1 score	21 (19–23)	18 (15–21)	<0.01
UPDRS part 2 score	35 (27–38)	26 (22–31)	<0.01
UPDRS part 3 (on state) score	35 (27–51)	25 (19–33)	<0.01
UPDRS part 3 (off state) score	65 (54–85)	57 (51–67)	0.01
UPDRS part 4 score	9 (6–11)	8 (6–10)	0.55
UDysRs score	0 (0–22)	0 (0–25)	0.94
NMSS score	19 (17–21)	17 (15–19)	<0.01
SSA score	21 (21–22)	21 (20–21)	<0.01
Kubota drinking test	20	94	0.74
KPPS score	8 (6–12)	7 (0–12)	0.30
HAMD score	13.5 (11–17)	12 (8–17)	0.10
HAMA score	10.5 (7–12)	9 (7–13)	0.46

Data are reported as frequency (percentage), or median (inter-quartile range). UPDRS, unified Parkinson’s disease rating scale; UDysRs, unified dyskinesia rating scale; NMSS, non-motor symptom scale; SSA, standardized swallowing assessment; KPPS, king’s parkinson’s disease pain scale; HAMD, hamilton depression scale; HAMA, hamilton anxiety scale; POD, postoperative delirium.

**Table 3 brainsci-13-00025-t003:** Anesthetics and postoperative brain CT imaging data.

	POD Group (*n* = 22)	Non-POD Group (*n* = 109)	*p*
Drugs			
Propofol (mg)	790 (640–900)	750 (620–850)	0.96
Sufentanil (µg)	40 (35–40)	40 (35–40)	0.74
Remifentanil (mg)	1.35 (1.2–1.6)	1.4 (1.2–1.6)	0.87
Rocuronium (mg)	50 (50–50)	50 (50–50)	0.89
Imaging Data			
Hemorrhage	10 (45.5)	20 (18.3)	<0.01
Edema	12 (54.5)	35 (32.1)	0.04
Pneumocephalus	11 (50.0)	59 (54.1)	0.723

Data are reported as median (inter-quartile range) or frequency (percentage). POD, postoperative delirium.

**Table 4 brainsci-13-00025-t004:** Multivariate logistic regression analysis showing the independent predictors of POD.

	OR	95% CI		*p*
		Lower	Upper	
Preoperative MMSE score	0.855	0.768	0.951	0.004
UPDRS part 3 (on state) score	1.061	1.02	1.104	0.003

MMSE, Mini-Mental State Examination; UPDRS, unified Parkinson’s disease rating scale; POD, postoperative delirium; OR, odds ratio; CI, confidence interval.

## Data Availability

The data supporting the findings of this study was obtained from the corresponding author according to reasonable request, and the corresponding author/s can be directly contacted for further inquiry.

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
