# Peer review of "Risk Factors for Delirium after Deep Brain Stimulation Surgery under Total Intravenous Anesthesia in Parkinson’s Disease Patients"

_brainsci, 2022, doi:10.3390/brainsci13010025_

Round 1

Reviewer 1 Report

In this manuscript, Li et al assessed factors predisposing to postoperative delirium following STN DBS surgery for PD, in a large single centre prospective case series. Delirium was assessed using the CAM-ICU (Confusion Assessment Method - Intensive Care Unit) which was performed twice daily for at least 7 days postoperatively. They also assessed a number of pre-operative and peri-operative factors to determine which factors predict the development of delirium.

Postoperative delirium was noted in 22 out of 131 patients. A number of pre and per-operative factors were associated with the development of postoperative delirium including, older age, diabetes, lower pre-operative MMSE, higher pre-operative UPDRS, haemorrhage on CT and oedema on CT. Multivariate regression analysis using UDPRS and pre-operative MMSE demonstrated that both these factors predispose to postoperative delirium.

The results of this study are not particularly novel as several studies, some of which are quoted in the manuscript, have already looked at factors predisposing to postoperative delirium in PD STN DBS patients. Nonetheless this is a large case series including fairly detailed clinical assessment. I'm not aware that other studies assessed all these variables. Therefore I found the study interesting and worthy of publication, although not particularly noteworthy. I have a few questions:-

1) Did the authors perform any correction for multiple comparisons? A large numer of t-tests were performed (more than 20). Thus if using a p value significance of 0.05 at least some of these results are likely to be just by chance. Perhaps Bonferroni correction or simply using a p value of 0.01 would be more accurate.

2) I would be interested to see a bit more information on the logistic regression. Specifically to know how much variance in postoperative delirium is explained by the two predictors used. The odds ratios suggest not a great deal but would be interested to see an R squared or similar statistic. 

3) The haemorrhage rates look fairly high in this case series (47 out of 131). Can the authors comment on this?

Author Response

Thanks for your comments concerning our manuscript entitled “Risk factors for delirium after deep brain stimulation surgery under total intravenous anesthesia in Parkinson’s disease patients” (brainsci-2063176) in “Brain Sciences”. We sincerely thank you for the valuable feedback that we have used to improve the quality of our manuscript. The reviewer comments are laid out below in normal font and specific concerns have been numbered. Our response is given in red text. In addition, all revisions to the manuscript have been marked up using the “Track Changes” function in the manuscript.

We look forward to hearing from you regarding our submission. We would be glad to respond to any further questions and comments that you may have.

Comment 1: Did the authors perform any correction for multiple comparisons? A large number of t-tests were performed (more than 20). Thus if using a p value significance of 0.05 at least some of these results are likely to be just by chance. Perhaps Bonferroni correction or simply using a p value of 0.01 would be more accurate.

Response 1: Thanks for your suggestion and we have adopted a p value significance of 0.01 for multiple comparisons between two groups. Which was described in Lines 145-146.

Comment 2: I would be interested to see a bit more information on the logistic regression. Specifically to know how much variance in postoperative delirium is explained by the two predictors used. The odds ratios suggest not a great deal but would be interested to see an R squared or similar statistic.

Response 2: For this logistic regression model, Cox-snell R2 is 0.159 and Nagelkerke R2 is 0.268. The area under receiver operating characteristic (ROC) curve for preoperative MMSE score combined with UPDRS part 3 (on state) score as a predictor of POD was 0.784 (0.675-0.893) (Lines 190-192). The ROC curve was shown in the Figure 2.

Comment 3: The haemorrhage rates look fairly high in this case series (47 out of 131). Can the authors comment on this?

Response 3: Certainly, the haemorrhage rate was higher than previous studies due to multiple postoperative CT examinations in our study. However, all of them did not receive additional treatment due to mild brain hemorrhage with less than 3 ml. Which was also described in Discussion (Lines 268-271).

Reviewer 2 Report

Dear Authors

It is an interesting article evaluating presence of post op delirium in patients undergoing DBS

Few clarifications are required

1. The pre-operative MMSE is very low in  the delirium group -indicative of dementia ( average MMSE was 23 with range starting from 18) - the cut off value in most areas is 24 for diagnosing dementia. Please mention if the scales used for literate and illiterate population were same and the normative MMSE in these populations.

Most DBS centres use the cut off of MMSE 24 to avoid STN DBS as the cognition may worsen in such cases, else they prefer to do GPi DBS. What was the indication for DBS in these patients.

2. The difference in electrolytes and blood glucose values pre-operatively between the two groups is very small and not clinically meaningful. The serum sodium and chloride values were normal and the range of values were similar. Can you please check the statistics again to verify the significance. Were these parameters used in the logistic regression analysis

3. The rate of hemorrhage post operatively is very high in both the groups - 45% in the delirium group and 18.3% in the non delirium group, this is way higher than those seen all over the world (especially with asleep DBS surgeries) - what was the reason for this rate. How was the hemorrhage diagnosed - CT or  MRI?.  Invading the lateral ventricle is avoided in all centres. How many of the patients had lateral ventricle invasion. Please mention the approach taken during the DBS surgery (stereotactic, with or without frame, any intra op imaging used, MER recording performed)

4. What was the rate of pneumocephalus in these patients

5. What was the duration of the delirium in these patients 

Round 2

Reviewer 2 Report

Corrections made are satisfactory